# An optical method for deriving the anterior and posterior curvatures of intraocular lenses using dual back-vertex power measurements

Damien Gatinel [1,2]*

1 Rothschild Foundation Hospital, Paris, France, 2 Abulcassis International University of Health Sciences, Rabat, Morocco

* gatinel@gmail.com

## Abstract

We present a theoretical framework to estimate the anterior and posterior radii of curvature of a thick intraocular lens (IOL) by measuring its back-vertex power in two orientations. Armed with the lens thickness, refractive index, and a potential axial offset $d$ from haptic angulation, one can determine the individual surface powers and, thus, the geometry of the implant. Using paraxial optics, we derive the back-vertex power in normal and flipped orientations. We consider two cases: $d = 0$ (no haptic-induced offset) and $d \neq 0$ (finite shift). In the $d = 0$ case, using standard paraxial relations (y–$\nu$ method), we obtain compact expressions that allow direct recovery of the surface powers from the dual back-vertex powers. For $d \neq 0$, the measured powers are first mapped back to the lens vertex (Eq 11), after which the same closed-form retrieval as for $d = 0$ applies. When $d = 0$, a closed-form solution yields the surface powers $(P_1, P_2)$ and radii $(R_{ia}, R_{ip})$. If the lens is shifted by $d$, we first correct to the vertex plane (Eq 11) and then apply the same closed-form relations. Though lens nominal power alone does not reveal geometry, our dual-orientation approach recovers how much power resides on each surface, benefiting thick-lens IOL power formulas and refining predictions in cataract surgery planning.

## Introduction

The IOL power alone does not sufficiently represent the refractive properties of the lens in the eye. To accurately depict the refraction of the lens in a pseudophakic eye, we must consider the curvatures or power values of both lens surfaces, or use the equivalent power along with the Coddington shape factor, the central thickness of the IOL, and the refractive index of the lens optic material. Modern intraocular lens (IOL) power calculation formulas increasingly adopt a thick-lens model for both the cornea and the implant, leveraging detailed geometric data to enhance the accuracy of postoperative refraction predictions [1–4]. Traditionally, IOLs are specified by a single nominal or equivalent power (e.g., 22 D), which obscures how the total power is

**Data availability statement:** All relevant data are within the paper.

**Funding:** The author(s) received no specific funding for this work.

**Competing interests:** The authors have declared that no competing interests exist.

distributed between the anterior ($R_{ia}$) and posterior ($R_{ip}$) surfaces, as well as the lens thickness.

While some manufacturers provide lens-shape data, such as the Coddington factor or front/back curvatures for specific power steps, this information is often unavailable. The IOL Power Club (https://www.iolpowerclub.org) has recently launched an initiative to encourage IOL manufacturers to share the fundamental design specifications of their intraocular lenses, enabling more accurate lens power calculations using a thick lens model for both the cornea and the lens [5]. Without knowing the individual surface powers, fully exploiting thick-lens models becomes challenging [6]. This raises the question of whether we can deduce an IOL's geometry through purely optical means, without relying on proprietary data.

In this study, we propose a theoretical framework to investigate whether measuring the back-vertex power in two orientations—normal and flipped—using a standard lens meter or optical bench can enable the deduction of the paraxial design of an intraocular lens (IOL), specifically its anterior and posterior curvatures. A single power measurement is insufficient to disentangle the contributions of the anterior and posterior surfaces, as the labeled IOL power alone does not reveal the distribution between these surfaces. We hypothesize that dual measurements, combined with the lens's thickness ($t$) and refractive index ($n_{IOL}$), could allow us to solve for the surface powers ($P_1, P_2$) and thus determine the curvatures ($R_{ia}, R_{ip}$).

Furthermore, we explore the theoretical impact of a potential axial offset ($d \neq 0$) caused by haptic angulation, which may shift the lens along the optical axis when flipped. Haptic angulation may create a small axial spacing $d$ between the lens back vertex and the detection plane in bench setups; we therefore treat $d$ as a measurable spacing and correct raw back-vertex readings to the vertex plane following the ISO 11979-2 procedure for vertex power measurement [7]. We aim to verify if accounting for this offset alters the back-vertex power calculations and requires numerical solutions.

If the lens thickness is unknown, we consider whether it can be estimated from nominal power, design trends, or manufacturer-provided thickness tables. Ultimately, we seek to confirm that the computed thick-lens power $P$ (the intrinsic power at the principal planes) aligns closely with the lens's labeled nominal power. Through this paraxial optical approach, we aim to provide a noninvasive method to support lens designers, surgeons, and researchers in refining thick-lens power formulas, particularly as IOL designs vary in shape factor and thickness across different power ratings.

## Materials and methods

### Theoretical framework

To develop the theoretical framework for measuring back-vertex power, we begin by defining the surface powers and the intrinsic power of the thick lens, which form the foundation of our optical model.

**Surface powers and thick-lens power.** Let:

- $n_{\text{IOL}}$: refractive index of the lens,
- $t$: axial thickness,
- $n_a$: index of the surrounding medium (often $\approx 1.0$ in air or 1.334 in aqueous),
- $R_{\text{ia}}$: anterior radius of curvature,
- $R_{\text{ip}}$: posterior radius of curvature (negative for biconvex in usual sign convention).

We define the surface powers:

$$P_1 = \frac{n_{\text{IOL}} - n_a}{R_{\text{ia}}}, \quad P_2 = \frac{n_a - n_{\text{IOL}}}{R_{\text{ip}}}, \tag{1}$$

and the intrinsic power $P$ of the thick lens (between principal planes):

$$P = P_1 + P_2 - \frac{t}{n_{\text{IOL}}}(P_1 P_2). \tag{2}$$

We will use R1 (anterior) and R2 (posterior) in tables and figures; $R1 \equiv R_{\text{ia}}$ and $R2 \equiv R_{\text{ip}}$.

**Back-vertex power and analytical solution ($d = 0$).**

**Setup.** Let $\tau = t/n_{\text{IOL}}$. Denote by $A$ and $B$ the measured back-vertex powers in the normal and flipped orientations, respectively. The surface powers are given by Eq 1 and the thick-lens power by Eq 2.

**Nomenclature and wavelength.** We denote by $A \equiv P_{\text{BV}}^{(\text{normal})}(d=0)$ and $B \equiv P_{\text{BV}}^{(\text{flipped})}(d=0)$ the back-vertex powers measured at the lens vertex in the two orientations. All vergences and material refractive indices are evaluated at the bench wavelength $\lambda$ (typically the mercury e-line at 546.07 nm), consistent with ISO 11979-2. [7] When a spacing $d$ is present between the lens back vertex and the measurement plane, we correct the readings (see "Offset correction") to equivalent vertex powers $A^\star(d)$ and $B^\star(d)$ prior to applying the $d=0$ relations.

**Compact y–$\nu$ derivation.** Choose unit input ray height $y_1 = 1$ and use reduced angle $\nu = n\,u$. In the normal orientation (surface $P_1$ first), the paraxial steps are

$$n_{\text{IOL}} u_1 = -y_1 P_1, \tag{3}$$
$$y_2 = y_1 + \tau\, n_{\text{IOL}} u_1 \ = \ 1 - \tau P_1, \tag{4}$$
$$u_2 = n_{\text{IOL}} u_1 - y_2 P_2 \ = \ -P_1 - P_2 + \tau P_1 P_2. \tag{5}$$

By definition of back-vertex power in this configuration,

$$\frac{1}{A} \ = \ -\frac{y_2}{u_2} \quad \Longleftrightarrow \quad u_2 = -A\, y_2. \tag{6}$$

Combining (4)–(6) gives

$$A\,(1 - \tau P_1) \ = \ P_1 + P_2 - \tau P_1 P_2 \ \equiv \ P. \tag{7}$$

Hence the linear relation

$$P_1 \ = \ \frac{1}{\tau}\left(1 - \frac{P}{A}\right) = \frac{n_{\text{IOL}}}{t}\left(1 - \frac{P}{A}\right). \tag{8}$$

Flipping the IOL (surface $P_2$ first) yields analogously

$$P_2 = \frac{1}{\tau}\left(1 - \frac{P}{B}\right) = \frac{n_{IOL}}{t}\left(1 - \frac{P}{B}\right). \tag{9}$$

**Closed-form retrieval.** Using (8)–(9) in the identity $P = P_1 + P_2 - \tau P_1 P_2$ (Eq 2) produces a short quadratic in $P$. Selecting the convergent root determines $P$, from which $(P_1, P_2)$ follow directly via (8)–(9). Finally, the radii are obtained from Eq 1.

The setup for measuring the back-vertex power in normal and flipped orientations, with and without axial offset due to haptic angulation, is illustrated in Fig 1.

**Offset correction for a measurement spacing ($d \neq 0$).** If the detection plane is located a distance $d$ downstream in medium $n_a$, the vergence updates as

$$V_{out} = \frac{V_{in}}{1 - \frac{d}{n_a}V_{in}}, \tag{10}$$

so that the *vertex* vergence is recovered by inversion

$$V_{in} = V_{out}/(1 + (d/n_a) * V_{out}). \tag{11}$$

We therefore correct raw bench readings $A_{meas}(d)$ and $B_{meas}(d)$ to the equivalent vertex powers

$$A^\star(d) = \frac{A_{meas}(d)}{1 + \frac{d}{n_a}A_{meas}(d)}, \qquad B^\star(d) = \frac{B_{meas}(d)}{1 + \frac{d}{n_a}B_{meas}(d)},$$

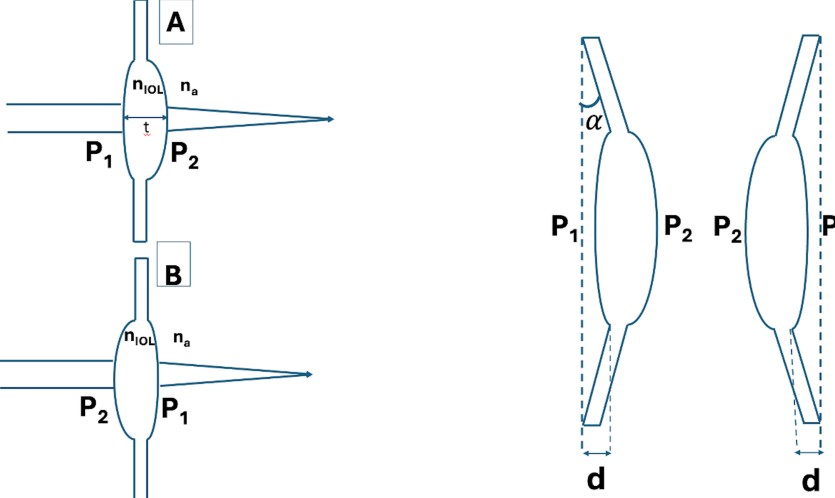

**Fig 1. Measurement of IOL back-vertex power.** Schematic representation of intraocular lens (IOL) power measurements in two orientations. A and B are measured back-vertex powers. A: Normal orientation ($d = 0$): collimated input, anterior surface $P_1$ then posterior $P_2$; measured vertex power A. B: Flipped orientation ($d = 0$): $P_2$ then $P_1$; measured vertex power B. (C) Configuration with a spacing $d$ between the back vertex and the detection plane (e.g., clamp geometry or effective haptic angulation $\alpha$); raw readings are corrected to vertex powers via Eq 11. Indices $n_a$ (ambient) and $n_{IOL}$ (optic) are indicated.

and then use the $d=0$ y–$\nu$ relations with $A \leftarrow A^\star(d)$ and $B \leftarrow B^\star(d)$. No additional algebra beyond Eqs 1–2 and the compact derivation is required. This correction maps the measured vergence at the detection plane back to the IOL vertex, consistent with the vertex-power definition in ISO 11979-2. [7]

A minimal Python implementation is available from the author on request.

## Results

### Numerical examples

**Case 1: $d = 0$.** Assume $n_{\mathrm{IOL}} = 1.46$, $t = 1.0\,\mathrm{mm}$, $n_a = 1.000$. Measuring the lens on an ISO 11979-2 compliant wet-cell optical bench (aqueous medium), we find:

$$A = 22.23\,\mathrm{D}\ (\text{normal}), \quad B = 22.35\,\mathrm{D}\ (\text{flipped}). \tag{12}$$

Solving the quadratic relation in $P$ implied by the y–$\nu$ relations (Methods) yields a unique positive root $P \approx 22.12\,\mathrm{D}$. Then from the partial-power formulas, we obtain $P_1 \approx 7.20\,\mathrm{D}$, $P_2 \approx 15.00\,\mathrm{D}$, and radii e.g. $R_{\mathrm{ia}} \approx +17.51\,\mathrm{mm}$, $R_{\mathrm{ip}} \approx -8.40\,\mathrm{mm}$.

**Case 2: Sensitivity to $d \neq 0$ and $t$.** To explore the impact of the axial offset $d$ (due to haptic angulation) and lens thickness $t$ on the IOL geometry, we analyzed three biconvex IOLs with different power levels: a low-power IOL ($A = 15.72\,\mathrm{D}$, $B = 15.79\,\mathrm{D}$, $t = 0.85\,\mathrm{mm}$), a medium-power IOL ($A = 24.9\,\mathrm{D}$, $B = 25.1\,\mathrm{D}$, $t = 1.15\,\mathrm{mm}$), and a high-power IOL ($A = 34.34\,\mathrm{D}$, $B = 33.71\,\mathrm{D}$, $t = 1.45\,\mathrm{mm}$). We used a modified version of a Python script, which ensures that only biconvex solutions ($P_1 > 0$, $P_2 > 0$) are retained, computing the anterior ($R_1$) and posterior ($R_2$) radii of curvature as $d$ varies from 0 to 1 mm (in steps of 0.1 mm) and as $t$ varies from $t_0 - 0.10\,\mathrm{mm}$ to $t_0 + 0.20\,\mathrm{mm}$ in 0.10-mm steps ($\Delta t \in \{-0.10, 0.00, +0.10, +0.20\}\,\mathrm{mm}$; $d = 0$). The script first corrects the measured powers to $A^\star(d), B^\star(d)$ via Eq 11 and then applies the $d=0$ y–$\nu$ relations (Methods) to retrieve ($P, P_1, P_2$). Fixed optical parameters were set as $n_{\mathrm{IOL}} = 1.46$ and $n_a = 1.334$ (aqueous medium).

For each analysis, we provide both a numerical table (for exact reproducibility) and a graphical figure (for visual appraisal of trends); the figures are plotted from the same data as the tables. Table 1 shows the influence of $d$ on $R_1$ and $R_2$. For the low-power IOL, as $d$ increases from 0 to 1 mm, $R_1$ increases from 31.058 mm to 31.425 mm (a 1.2%

**Table 1**. **Influence of axial offset $d$ on radii of curvature.** Numerical results showing the effect of varying axial offset $d$ from 0 to 1 mm on the anterior ($R_1$) and posterior ($R_2$) radii of curvature for three IOLs: low-power ($A = 15.72\,\mathrm{D}$, $B = 15.79\,\mathrm{D}$, $t = 0.85\,\mathrm{mm}$), medium-power ($A = 24.9\,\mathrm{D}$, $B = 25.1\,\mathrm{D}$, $t = 1.15\,\mathrm{mm}$), and high-power ($A = 34.34\,\mathrm{D}$, $B = 33.71\,\mathrm{D}$, $t = 1.45\,\mathrm{mm}$). Intrinsic powers from the y–$\nu$ solution (at $d=0$) were $P \approx 15.68\,\mathrm{D}$ (low), $P \approx 24.76\,\mathrm{D}$ (medium), and $P \approx 33.45\,\mathrm{D}$ (high).

| | Low-Power IOL | | Medium-Power IOL | | High-Power IOL | |
|---|---|---|---|---|---|---|
| $d$ (mm) | $R_1$ (mm) | $R_2$ (mm) | $R_1$ (mm) | $R_2$ (mm) | $R_1$ (mm) | $R_2$ (mm) |
| 0.00 | 31.058 | -10.812 | 17.028 | -7.218 | 4.848 | -16.450 |
| 0.10 | 31.095 | -10.825 | 17.059 | -7.231 | 4.860 | -16.492 |
| 0.20 | 31.132 | -10.838 | 17.091 | -7.245 | 4.872 | -16.533 |
| 0.30 | 31.168 | -10.850 | 17.123 | -7.258 | 4.884 | -16.575 |
| 0.40 | 31.205 | -10.863 | 17.155 | -7.272 | 4.897 | -16.617 |
| 0.50 | 31.242 | -10.876 | 17.187 | -7.285 | 4.909 | -16.659 |
| 0.60 | 31.278 | -10.889 | 17.218 | -7.298 | 4.921 | -16.701 |
| 0.70 | 31.315 | -10.901 | 17.250 | -7.312 | 4.933 | -16.742 |
| 0.80 | 31.352 | -10.914 | 17.282 | -7.325 | 4.945 | -16.784 |
| 0.90 | 31.388 | -10.927 | 17.314 | -7.339 | 4.958 | -16.826 |
| 1.00 | 31.425 | -10.939 | 17.346 | -7.352 | 4.970 | -16.868 |

Computed using an internal Python script (available on request), modified to iterate over $d$ values, ensuring biconvex solutions.

change), and $R_2$ decreases from -10.812 mm to -10.939 mm (a 1.2% change). The medium-power IOL shows $R_1$ increasing from 17.028 mm to 17.346 mm (1.9%) and $R_2$ from -7.218 mm to -7.352 mm (1.9%). The high-power IOL exhibits a more pronounced effect, with $R_1$ increasing from 4.848 mm to 4.970 mm (2.5%) and $R_2$ from -16.450 mm to -16.868 mm (2.5%). Table 2 examines the effect of varying $t$. These thickness-dependent trends are visualized in Fig 3, plotted from the same dataset as Table 2. For the low-power IOL, as $t$ changes from 0.75 mm to 1.05 mm, $R_1$ decreases from 35.496 mm to 26.359 mm (25.8% decrease), and $R_2$ decreases from -10.359 mm to -11.535 mm (11.4% decrease). These thickness-dependent trends are visualized in Fig 3, plotted from the same dataset as Table 2. The medium-power IOL shows $R_1$ decreasing from 18.207 mm to 15.474 mm (15.0%) and $R_2$ from -7.022 mm to -7.547 mm (7.5%). The high-power IOL has $R_1$ increasing from 4.721 mm to 5.071 mm (7.4%) and $R_2$ increasing from -18.062 mm to -14.359 mm (20.5% decrease in magnitude).

**Table 2**. **Influence of lens thickness $t$ on radii of curvature.** Numerical results showing the effect of varying lens thickness $t$ over $\Delta t \in \{-0.10, 0.00, +0.10, +0.20\}$ mm around the nominal value $t_0$ ($d = 0$) on the anterior ($R_1$) and posterior ($R_2$) radii of curvature for three IOLs: low-power ($t_0 = 0.85$ mm), medium-power ($t_0 = 1.15$ mm), and high-power ($t_0 = 1.45$ mm). Measured powers are the same as in Table 1. Intrinsic powers from the y–v solution (at $d=0$) were $P \approx 15.68$ D (low), $P \approx 24.76$ D (medium), and $P \approx 33.45$ D (high).

| $\Delta t$ (mm) | Low-Power IOL | | Medium-Power IOL | | High-Power IOL | |
|---|---|---|---|---|---|---|
| | $R_1$ (mm) | $R_2$ (mm) | $R_1$ (mm) | $R_2$ (mm) | $R_1$ (mm) | $R_2$ (mm) |
| -0.10 | 35.496 | -10.359 | 18.207 | -7.022 | 4.721 | -18.062 |
| +0.00 | 31.058 | -10.812 | 17.028 | -7.218 | 4.848 | -16.450 |
| +0.10 | 28.271 | -11.200 | 16.151 | -7.392 | 4.964 | -15.266 |
| +0.20 | 26.359 | -11.535 | 15.474 | -7.547 | 5.071 | -14.359 |

Computed using an internal Python script (available on request), modified to iterate over $\Delta t$ values, ensuring biconvex solutions.

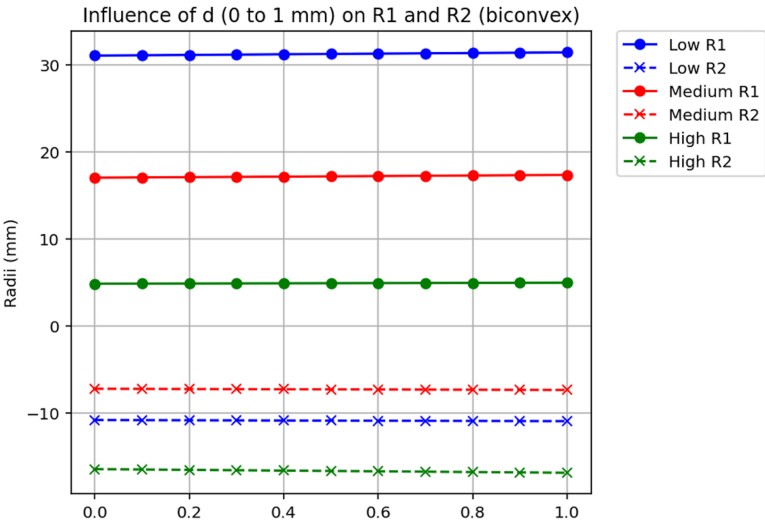

**Fig 2**. **Effect of axial spacing $d$ on recovered radii.** Radii of curvature for the anterior surface ($R_1$) and the magnitude of the posterior radius ($|R_2|$) are plotted as functions of the vertex–detector axial spacing $d \in [0, 1.0]$ mm in 0.10-mm steps, for three biconvex IOLs: low-power ($A = 15.72$ D, $B = 15.79$ D, $t = 0.85$ mm), medium-power ($A = 24.9$ D, $B = 25.1$ D, $t = 1.15$ mm), and high-power ($A = 34.34$ D, $B = 33.71$ D, $t = 1.45$ mm). Fixed optical parameters were $n_{IOL} = 1.46$ and $n_a = 1.334$. A biconvex constraint ($P_1 > 0$, $P_2 > 0$) was enforced. For plotting clarity the absolute value $|R_2|$ is shown; radii are signed in the text/tables. Curves are generated from the same dataset reported in Table 1.

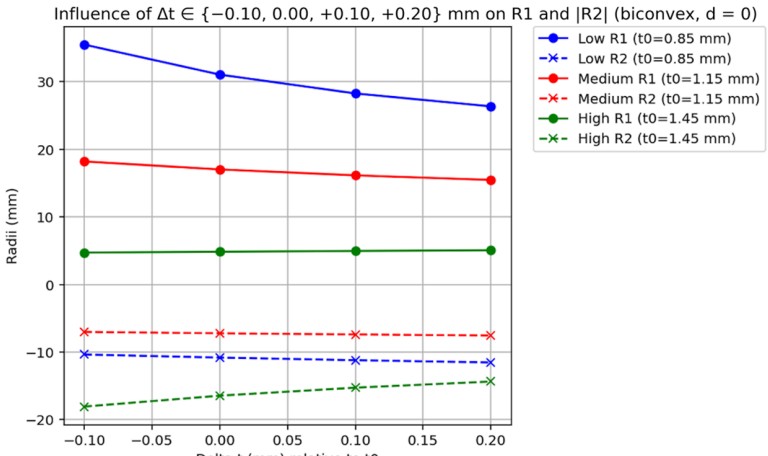

**Fig 3**. **Effect of thickness $t$ on recovered radii at $d = 0$.** Radii of curvature for the anterior surface ($R_1$) and the magnitude of the posterior radius ($|R_2|$) are plotted for thickness values $t = t_0 + \Delta t$ with $\Delta t \in \{-0.10, 0.00, +0.10, +0.20\}$ mm, where $t_0 = 0.85$ mm (low-power IOL), 1.15 mm (medium-power), and 1.45 mm (high-power). The measured vertex powers are the same as in Fig. 2/Table 1 (low: $A = 15.72$ D, $B = 15.79$ D; medium: $A = 24.9$ D, $B = 25.1$ D; high: $A = 34.34$ D, $B = 33.71$ D), with $n_{IOL} = 1.46$ and $n_a = 1.334$; $d = 0$ for all points. A biconvex constraint ($P_1 > 0$, $P_2 > 0$) was enforced. For plotting clarity $|R_2|$ is shown; radii are signed in the text/tables. Curves are generated from the same dataset reported in Table 2.

## Discussion

Our method aims to deduce the geometry (anterior vs. posterior power) of a thick IOL from purely optical power measurements in two orientations. This is important because many advanced lens-calculation formulas—especially ones that treat the cornea and IOL as thick lenses—can exploit such geometric data. A study has presented the outcomes of a Monte-Carlo simulation utilizing a large dataset of biometric measurements obtained from a modern optical biometer, employing linear Gaussian optics techniques to examine the impact of modeling the intraocular lens (IOL) as a thick lens rather than a thin lens on spherical equivalent refraction and ocular magnification [8]. The findings indicate that adopting a more accurate thick lens model for the IOL (with a Coddington factor ranging from -1.0 to 1.0) instead of a thin lens model (with the same IOL power and axial position) led to a variation in spectacle refraction of up to ±1.5 diopters. If we do not measure the lens in both orientations, we cannot separate front-surface from back-surface power.

A study by Hoffer et al. (2009), conducted at the U.S. Food and Drug Administration Optical Testing Laboratory, evaluated the precision of intraocular lenses (IOLs) labeled with exact dioptric power using a confocal laser method [9]. This method, compliant with ISO 11979-2 and ANSI Z80.7 standards, demonstrated exceptional repeatability, with standard deviations below ±0.02 D for positive IOLs and ±0.01 D for negative IOLs, and relative errors less than 0.05%. These results indicate a precision far superior to the ISO/ANSI tolerances, which range from ±0.30 D to ±1.00 D depending on the IOL power. The study found an average difference of 0.18 D (±0.12 SD) between the manufacturer's exact labeled power and the measured power, compared to 0.23 D (±0.09 SD) between the measured power and the conventionally rounded labeled power (in 0.50 D steps), yielding an improvement of 0.05 D with exact labeling. This improvement was more pronounced for IOLs in the 15.00 D to 20.00 D range, with an average difference of 0.08 D (±0.05 SD) between the exact labeled and measured power versus 0.17 D (±0.06 SD) for the rounded labeled power, resulting in a statistically significant improvement of 0.09 D ($p = 0.008$). However, for IOLs above 20.00 D, the improvement was minimal at 0.03 D and not statistically significant ($p = 0.448$). Notably, these differences between labeled and measured power do not pose a concern for the method proposed in this study, as it does not rely on the labeled power but instead estimates the IOL power directly from back-vertex power measurements in both normal and flipped orientations, thereby mitigating potential discrepancies introduced by labeling inaccuracies.

Knowledge of the IOL design is particularly critical for high-power implants, as the labeled power is typically defined relative to the image-side principal plane of the implant. For a biconvex IOL, this principal plane is located between the anterior surface (in a nearly convex-plano configuration) and the posterior surface (in a nearly plano-convex configuration). In low-power implants, the uncertainty associated with the principal plane position is more constrained due to their thinner profile, as demonstrated by Gatinel et al. (2022) in a theoretical thick lens model [10]. The quadratic dependence on IOL power underscores that the impact of ELP variations on postoperative refraction increases significantly with higher IOL powers, making the IOL design a more critical factor for accurate power calculations in such cases [11]. Accurate thick-lens parameters are most impactful when lens power and thickness are high, because the location of the principal planes and ELP-to-refraction coupling increase with $P$. Our contribution is limited to retrieving $(P_1, P_2)$ (and thus the Coddington shape) from dual back-vertex measurements; how this information should be incorporated into a given formula or ray-tracing pipeline is outside our scope.

The numerical results from our theoretical model provide further insight into the sensitivity of the calculated anterior ($R_1$) and posterior ($R_2$) radii of curvature to variations in axial offset ($d$) and lens thickness ($t$). For a low-power IOL ($A = 15.72$ D, $B = 15.79$ D, $t = 0.85$ mm), the anterior radius $R_1$ increased from 31.058 mm to 31.425 mm, and the posterior radius $R_2$ decreased from -10.812 mm to -10.939 mm as $d$ varied from 0 to 1 mm, indicating a relatively modest change (approximately 1.2% for $R_1$ and 1.2% for $R_2$). In contrast, for a high-power IOL ($A = 34.34$ D, $B = 33.71$ D, $t = 1.45$ mm), the same variation in $d$ resulted in $R_1$ increasing from 4.848 mm to 4.970 mm (a 2.5% change) and $R_2$ decreasing from -16.450 mm to -16.868 mm (a 2.5% change), demonstrating a more pronounced sensitivity. Similarly, when varying the thickness over $\Delta t \in \{-0.10, 0.00, +0.10, +0.20\}$ mm around the nominal value, the low-power IOL exhibited changes in $R_1$ from 35.496 mm to 26.359 mm (a 25.8% decrease) and in $R_2$ from -10.359 mm to -11.535 mm (an 11.4% decrease), while the high-power IOL showed changes in $R_1$ from 4.721 mm to 5.071 mm (a 7.4% increase) and in $R_2$ from -18.062 mm to -14.359 mm (a 20.5% increase). These results align with the theoretical expectation that variations in ELP and thickness have a greater relative impact on high-power IOLs due to the quadratic relationship with IOL power [11]. For low-power IOLs, the smaller relative changes in radii suggest that the distribution of power between the anterior and posterior surfaces is less critical, as the thinner profile and lower power reduce the impact of such variations on postoperative refraction. Conversely, for high-power IOLs, precise knowledge of the IOL design becomes increasingly important, as even small changes in ELP or thickness can lead to significant refractive errors. The precision of back-vertex power measurements, as demonstrated by Hoffer et al. (2009) with standard deviations below 0.02 D, ensures that our method can reliably capture these variations, enabling accurate estimation of $R_1$ and $R_2$ across a range of IOL powers [9].

The proposed method is fully consistent with geometric optics under the paraxial approximation. While this study focuses on the theoretical feasibility of this approach, the principles of geometric optics suggest that methods validated theoretically are often readily translatable to practical applications, provided the measurement techniques are implementable. In this case, the use of standard lens meters or optical benches ensures that the proposed method is practically feasible, paving the way for future experimental validation to confirm its accuracy in real-world settings.

Recent clinical work in multifocal IOLs reported that, with optimized constants, a thick-lens formula showed no clinically relevant prediction-error advantage over Barrett in the 18–27 D range, and using exact versus labeled powers did not materially change accuracy in that cohort. [12] However, even if such constant adjustment can effectively zeroize the mean prediction error across a population, this approach does not prevent potentially large individual errors, particularly in high-power lenses where design-related variability has a stronger clinical impact. In this context, our method is complementary: by recovering $(P_1, P_2)$ and the shape factor from dual back-vertex measurements, it provides geometry that can help mitigate such individual risks.

## Assumptions and limitations

The primary assumption underlying this method is the paraxial approximation, which requires small angles and rays near the optical axis. The formula for back-vertex power remains valid as long as the lens is not excessively thick or powerful, or significantly tilted. Additionally, the equation for additional propagation, $d/n_a$, assumes a pure axial translation of the lens, without significant decentering or tilt. Notably, the influence of $d$ on these measurements is relatively small, which allows the analytical solution developed for the no-offset case to be applied to angulated implants as a first approximation, simplifying initial calculations before considering numerical refinements.

For IOLs with haptic angulation of approximately 5 °-15 ° [13],[14], the center of the lens may shift by approximately $3\,\text{mm} \times \sin(\text{angle}) \approx 0.25 - 0.8\,\text{mm}$ (Here, 3 mm corresponds to the half-diameter of a 6-mm optic.). We incorporate that offset $d$ to refine the back-vertex power equations [Eq 11]. Numerically solving the resulting system yields the same final output $(P, P_1, P_2)$.

Some manufacturers provide thickness or shape factor data for each lens power step, but many do not. If the thickness is unknown, one might infer it from the nominal lens power plus typical design guidelines. In any case, after we solve for $P$, the result generally lies close to the lens's labeled power (e.g., 22 D, which is referenced to the image-side principal plane according to the ISO standard).

Because lens design sometimes varies with power (for example, the posterior radius changes more rapidly than the anterior), knowledge of $(R_{ia}, R_{ip})$ can help identify whether the lens at 20 D is plano-convex or bi-convex, which is particularly significant for advanced modeling in borderline or post-refractive corneas. In such corneas, altered geometry, such as irregular curvature in borderline cases (e.g., early keratoconus or marginal pellucid degeneration) or changes in the anterior-posterior curvature ratio after refractive surgery (e.g., LASIK, PRK)—leads to challenges in predicting effective lens position (ELP) and managing higher-order aberrations. A plano-convex IOL, with one flat surface, positions its principal plane differently compared to a biconvex IOL, affecting the ELP and thus the postoperative refraction. Advanced modeling techniques, such as ray tracing, require precise knowledge of the surface curvatures of the IOL to optimize refractive results in these non-standard corneas, where small errors can lead to significant visual distortion or refractive surprises, particularly for patients with high expectations of visual quality [14].

**Refractive index and wavelength.** The method assumes $n_{IOL}$ and $n_a$ specified at the bench wavelength (often 546.07 nm). Because $R_{1,2} \propto (n_{IOL} - n_a)/P_{1,2}$, small index errors map nearly linearly to radii. For the three examples here, changing $n_{IOL}$ by $\pm 0.001$ ( 0.07%) changes $R_1$ and $R_2$ by about $\pm(0.8\text{–}0.9)\%$ at fixed $A, B, t$. This highlights the benefit of using material data (or Abbe-number dispersion) at the measurement $\lambda$.

**Lower-order aberrations and alignment.** Our paraxial model assumes coaxial surfaces. IOL tilt and decentration can induce defocus/astigmatism at the detector and bias the back-vertex reading; careful centration/tilt control is standard in bench methods and ISO 11979-2. Clinically, modest tilt/decentration in modern monofocal designs (e.g., TECNIS ZCB00) tends not to degrade VA substantially, but they can still perturb bench-measured vergences at the $10^{-2}$–$10^{-1}$ D level if uncorrected. [15]

## Conclusion

We have presented an optical procedure to separate the total power of a thick IOL into anterior and posterior components. By measuring two back-vertex powers (normal versus flipped) and knowing the thickness, refractive index, and any axial offset $d$, we solve for $(P_1, P_2)$ and thus $(R_{ia}, R_{ip})$. This method is purely noninvasive and can be implemented with standard lens-meter instruments. Our aim is to empower more precise thick-lens IOL power calculations and expand geometry data, which can be especially useful when lens design changes the shape factor across different nominal powers.

## Author contributions

**Conceptualization:** Damien Gatinel.

**Formal analysis:** Damien Gatinel.

**Methodology:** Damien Gatinel.

**Writing – original draft:** Damien Gatinel.

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
