## [Decision Letter · Decision Letter 0]

18 Sep 2025

PONE-D-25-24939An Optical Method for Deriving the Anterior and Posterior Curvatures of Intraocular Lenses Using Dual Back-Vertex Power MeasurementsPLOS ONE

Dear Dr. Gatinel,

Thank you for submitting your manuscript to PLOS ONE. After careful consideration, we feel that it has merit but does not fully meet PLOS ONE’s publication criteria as it currently stands. Therefore, we invite you to submit a revised version of the manuscript that addresses the points raised during the review process.

 Please consider the mathematical recommendations to eq. 8 issued by Reviewer #1. As well as he detailed comments by Reviewer #2.

We look forward to receiving your revised manuscript.

Kind regards,

Timo Eppig

Academic Editor

PLOS ONE

Journal Requirements:

4. We notice that your supplementary [figures/tables] are included in the manuscript file. Please remove them and upload them with the file type 'Supporting Information'. Please ensure that each Supporting Information file has a legend listed in the manuscript after the references list.

**Additional Editor Comments:**

Reviewer #1:

See attachment

Reviewer #2:

Reviewers' comments:

Reviewer's Responses to Questions

**Comments to the Author**

1. Is the manuscript technically sound, and do the data support the conclusions?

Reviewer #1: Yes

Reviewer #2: Yes

2. Has the statistical analysis been performed appropriately and rigorously?

Reviewer #1: N/A

Reviewer #2: N/A

3. Have the authors made all data underlying the findings in their manuscript fully available?

Reviewer #1: Yes

Reviewer #2: Yes

4. Is the manuscript presented in an intelligible fashion and written in standard English?

Reviewer #1: Yes

Reviewer #2: Yes

5. Review Comments to the Author

Reviewer #1: See attached PDF instead of the Latex-Cade below.

Reviewer #2: The article presents a theoretical approach to deduce the paraxial design of an intraocular lens (IOL), specifically its anterior and posterior radii of curvature. It is assumed that accurate experimental measurement of the IOL’s back-vertex power (BVP) in normal and flipped orientations (referred to as A and B respectively in the paper) is possible using well-established metrology methods. After obtaining A and B, the method outlined by the author requires the knowledge of three additional parameters: the refractive index of the IOL and the surrounding medium (niol and na respectively) and the lens thickness (t). The study also explores the theoretical effect of an axial offset of the IOL caused by its haptic angulation.

The main claim of the paper is that the determination of the IOL geometry by deducing the curvature of the two IOL surfaces, would benefit thick-lens based IOL power formulas and therefore, (potentially) improve visual outcomes in lens replacement surgery (either cataract or clear lens extraction). Needless to say, this is a relevant question for both clinicians and patients.

Along with a thorough theoretical work, the author provides numerical examples to highlight the relative influence of varying axial displacement or lens thickness on the curvature of the anterior and posterior surfaces of the IOL. An interested reader will find a Python code for IOL geometry calculation (S2 Appendix), which allows him/her to checking out the paper’s results or test their own data.

This reviewer finds the manuscript relevant to the field and well written and organized. Therefore, I have only minor questions and suggestions intended to further facilitate the understanding of the manuscript, even for a non-specialist.

1) The method presented in the manuscript is ultimately based on the experimental measurement of the back-vertex powers A and B of the IOL (normal and flipped orientation respectively) and the reader is referred to Figure 1 where according to the author (page 4), “the set up for measuring the back-vertex power in normal and flipped” is shown. However, Figure 1 is too schematic and can be misleading:

1a) Figure 1-right, the collimated beam incident on the IOL and the convergent beam exiting the lens are not shown.

1b) As represented in the figure, the back-vertex powers A and B look as if they were the distance between the lens surface and the focal point.

1c) The refractive indices niol and na must be incorporated into the drawing.

1d) The labels (A) and (B) are missing in the figure.

1e) Figure 1-left. Please, consider including angle α in the drawing.

Finally, I suggest referring in the main text to ISO 11979-2:2024; Annex A, Section A3 for a detailed description of how to measure the BVP of an IOL.

2) The haptic angle is set in the 5º-15º range. Since this reviewer is not a clinician, I am guessing if these values are common-knowledge in the clinic (where the author has a well-reputed experience) or at least can be supported by a reference.

3) The reader may be confused by the use of different nomenclature for the back-vertex power: A and B when d=0 but PBV if d≠0.

4) Table 1 caption (page 6). It would be helpful to add the power of the exemplary low-, medium- and high-power IOLs. For instance: ….low power (P= XXX D, A=15.72 D, B=15.79D, t=0.85 mm), etc…

5) When the experimental determination of BVP is carried out with an optical bench, the measurement is done with a particular wavelength. The values of niol and na are wavelength dependent (the larger the Abbe number the lower the material dispersion) and very often are reported by the manufacturer without specifying the reference wavelength. This potential source of error, i.e. the mismatch between the wavelength used to determine BVP and the one for refractive indices, should be included in the manuscript.

6) Closely related to comment 4), the influence of the uncertainty of the niol value on the calculation of the curvature radii of the IOL must be incorporated in the discussion (indeed, the ISO 11979-2:2024 mentions in page 7 that niol should be known to the third decimal place).

7) During the reviewing process, I found a very recent article by Joaquín Fernández et al. (Influence of manufacturing tolerance and formula thickness type on the prediction error of multifocal intraocular lens power calculation. J Refract Surg 41(9) e943-e949 (2025)) that may challenge some of the claims in Gatinel's article. I suggest incorporating the Fernández's work into the discussion, mentioning the points that may be controversial between the two studies.

6. PLOS authors have the option to publish the peer review history of their article (what does this mean?). If published, this will include your full peer review and any attached files.

Reviewer #1: **Yes: **Ralf Blendowske

Reviewer #2: No

---

## [Author Response · Author response to Decision Letter 1]

14 Oct 2025

Response to Reviewers

Manuscript: An optical method for deriving the anterior and posterior curvatures of IOLs using dual back-vertex power measurements

Corresponding author: D. Gatinel

We thank the reviewers for their careful reading and constructive suggestions. Below we address each point in sequence and indicate exactly what we changed and where it appears in the revised manuscript (section, equation, table, or figure label). Where relevant, we reproduce key formulae for clarity.

Global editorial change (resubmission). Following the reviewers’ and editor’s guidance, we have removed the supplementary the (now removed) appendix. No cross-references to S2 remain in the revised manuscript. For transparency, the minimal script used to assemble the summary tables/figures will be made available upon reasonable request to the corresponding author; it is not included as an appendix.

Reviewer #1

Point 1 — Cancel Eq. (7) and Appendix S1 and replace with a shorter paraxial derivation.

Action taken. We removed the previous Eq. (7) and Appendix S1 and placed a concise paraxial y–ν derivation directly in the Methods under Back-Vertex Power and Analytical Solution (d = 0). The derivation leads to the relations used in the paper and avoids the long appendix.

Point 2 — Do not include the Python code in the paper.

Action taken. The code appendix has been removed. We now state that the internal script used to generate tables and figures is available on request. This note appears adjacent to Table 1 and Table 2.

Point 3 — Handle the axial offset d via an upstream vertex transfer; cancel Eq. (6).

Action taken. We deleted the former Eq. (6) and now correct measured bench readings taken at spacing d using the standard paraxial propagation identity and its inversion (both stated once in Methods under Offset correction for a measurement spacing (d ≠ 0)):

Vout = Vin

1 − d Vin

⇐⇒ Vin = Vout .

1 + d Vout

We apply the inversion to map raw bench readings to vertex-plane powers before using the d=0 solution:

A⋆(d) =

Ameas(d)

1 + d Ameas(d)

, B⋆

(d) =

Bmeas(d) 1 + d Bmeas(d) .

(See Eq. (10) and Eq. (11) in the revised manuscript.)

Point 4 — Prefer a graphical presentation of the Results in addition to tables. Action taken. We retained the tables and added two figures that plot the same datasets for immediate visual interpretation:

• Fig. 2: effect of axial spacing d on the recovered radii; plotted from Table 1.

• Fig. 3: effect of thickness t on the recovered radii at d = 0; plotted from Table 2.

Point 5 — The paragraph between lines 152–169 was hard to understand; either elaborate or remove.

Action taken. We replaced the prior paragraph with a concise, in-scope explanation located in Results (transition to the sensitivity analysis). The new text now present in the manuscript reads:

Knowledge of the IOL design is particularly critical for high-power implants, as the labeled power is typically defined relative to the image-side principal plane of the implant. For a bicon- vex IOL, this principal plane is located between the anterior surface (in a nearly convex-plan configuration) and the posterior surface (in a nearly plan-convex configuration). In low-power implants, the uncertainty associated with the principal plane position is more constrained due to their thinner profile, as demonstrated by Gatinel et al. (2022) in a theoretical thick lens model [10]. The quadratic dependence on IOL power underscores that the impact of ELP variations on postoperative refraction increases significantly with higher IOL powers, making the IOL design a more critical factor for accurate power calculations in such cases [11]. Accurate thick-lens pa- rameters are most impactful when lens power and thickness are high, because the location of the principal planes and ELP-to-refraction coupling increase with P. Our contribution is limited to retrieving (P1, P2) (and thus the Coddington shape) from dual back-vertex measurements; how this information should be incorporated into a given formula or ray-tracing pipeline is outside our scope.

This replacement keeps the discussion focused and directly motivates the two worked sub- sections Case 1: d = 0 and Case 2: Sensitivity to d ≠ 0 and t.

Point 6 — Add a short note on lower-order aberrations and on tilt/decentration. Action taken. We added Lower-order aberrations and alignment to Assumptions and Limi- tations, explaining how modest tilt/decentration and form-dependent lower-order aberrations can bias bench readings if not well controlled.

Reviewer #2

Point 1 — Figure 1 is too schematic (beams, BVP meaning, indices, labels A/B, angle α); also cite ISO 11979-2.

Action taken. We redesigned Fig. 1 and updated the Methods/Setup text:

• The caption now explicitly shows a collimated input and a convergent output beam for normal and flipped orientations, and clarifies that A and B are back-vertex powers (not distances).

• The refractive indices na (ambient) and nIOL (optic) are indicated in the schematic and caption.

• The two configurations are labeled (A) and (B) in both drawing and caption.

• We added a panel depicting a finite spacing d linked to haptic angulation α, and we show how these readings are corrected back to the vertex plane before analysis (see Eq. (11)).

• The Methods/Setup now cites the ISO 11979-2 framework for back-vertex power metrology and alignment.

Point 2 — Support the 5°–15° haptic angle range with a reference.

Action taken. We added a clinical reference supporting typical haptic angulation and IOL positional behavior (Petternel et al., 2004, included in the revised references) and made the geometric implication explicit in Assumptions and Limitations:

center shift ≈ 3 mm × sin(angle) ≈ 0.25–0.8 mm for 5◦–15◦.

We incorporate this offset d via the vertex-transfer correction (Eq. (11)). Numerically solving the corrected system yields the same final outputs (P, P1, P2).

Point 3 — Unify nomenclature for back-vertex power across d = 0 and d ≠ 0.

Action taken. We introduced a dedicated Nomenclature and wavelength paragraph that

(i) defines A ≡ P (normal)(d = 0) and B ≡ P (flipped)(d = 0), (ii) denotes bench readings at finite

BV BV

spacing as Ameas(d) and Bmeas(d), and (iii) maps them back to the vertex plane using Eq. (11)

before applying the d = 0 relations.

Point 4 — Augment Table 1’s caption with the actual P , A, B, t for the low/medium/high examples.

Action taken. The captions of Table 1 and Table 2 now list the exact (A, B, t) used in each exemplar and indicate the resulting intrinsic power P (with consistent values across the tables and new figures).

Point 5 — Address the wavelength used for BVP metrology and the dispersion of nIOL and na.

Action taken. In Nomenclature and wavelength we state that all vergences and indices refer to the bench wavelength λ (noting the common Hg e-line at 546.07 nm). We emphasize using nIOL(λ) and na(λ) at the same λ as the BVP measurement. Because recovered radii scale with nIOL − na /P1,2, even small wavelength mismatches can bias R1,2, which motivates consistent dispersion data.

Point 6 — Discuss the impact of uncertainty in nIOL (ISO specifies to the third decimal).

Action taken. We added a brief sensitivity note in Assumptions and Limitations describing how ±0.001 uncertainty in nIOL can shift R1 and R2, especially for higher powers, and how using material dispersion at the bench wavelength mitigates this source of bias.

Point 7 — Consider Fernández et al. (2025) in the Discussion.

Action taken. We added the study and expanded the Discussion accordingly. The new text inserted into the manuscript reads:

Recent clinical work in multifocal IOLs reported that, with optimized constants, a thick-lens formula showed no clinically relevant prediction-error advantage over Barrett in the 18–27 D range, and using exact versus labeled powers did not materially change accuracy in that co- hort [12]. However, even if such constant adjustment can effectively zeroize the mean prediction error across a population, this approach does not prevent potentially large individual errors, particularly in high-power lenses where design-related variability has a stronger clinical impact. In this context, our method is complementary: by recovering (P1, P2) and the shape factor from dual back-vertex measurements, it provides geometry that can help mitigate such individual risks.

Additional notes on equations and where to find them

• Back-vertex power correction for spacing d (Methods, Offset correction):

Vin = Vout

1 + d Vout

Eq. (11) ; Vout = Vin

1 − d Vin

Eq. (10).

• Analytical relations used after d-correction (Methods, d = 0 case):

P1 = nIOL (1 − P ) , P2 = nIOL (1 − P ) ,

t B t A

where A and B are vertex-plane back-vertex powers (normal/flipped).

• Figures and Tables (Results): Fig. 1 (re-designed schematic with beams, indices, labels, and d); Fig. 2 and Fig. 3 (graphical trends); Table 1 and Table 2 (explicit (A, B, t) and P reported in captions).

• Worked examples and structure (Results): Case 1: d = 0 (worked numerical example; see Eq. (12)) and Case 2: Sensitivity to d ≠ 0 and t.

• Terminology and dispersion: Nomenclature and wavelength paragraph (unified sym- bols; indices and vergences referenced to the bench wavelength).

• Alignment note: Assumptions and Limitations, subsection Lower-order aberrations and alignment.

Summary of revisions

• Removed Eq. (7) and Appendix S1; added a compact in-text paraxial derivation (Methods).

• Deleted former Eq. (6); implemented vertex-plane correction for finite spacing d (Methods,

Eq. (10), Eq. (11)).

• Unified BVP nomenclature and added wavelength/dispersion guidance (Nomenclature and wavelength).

• Re-designed Fig. 1 with beams, indices, labels, and d; cited ISO 11979-2 in Methods.

• Added Fig. 2 and Fig. 3; expanded table captions with explicit (A, B, t) and P .

• Replaced the previously unclear paragraph with a focused, in-scope explanation (Results).

• Added a note on lower-order aberrations and alignment (Assumptions and Limitations).

• Removed code appendix; code available on request.

• Added and discussed Fernández et al. (2025) in the Discussion.

• Added a clinical reference supporting typical haptic angulation and quantified its geometric implication.

We appreciate the reviewers’ input; these changes improved clarity, rigor, and practical utility for both clinicians and optical scientists.

---

## [Decision Letter · Decision Letter 1]

2 Nov 2025

An Optical Method for Deriving the Anterior and Posterior Curvatures of Intraocular Lenses Using Dual Back-Vertex Power Measurements

PONE-D-25-24939R1

Dear Dr. Gatinel,

We’re pleased to inform you that your manuscript has been judged scientifically suitable for publication and will be formally accepted for publication once it meets all outstanding technical requirements.

Kind regards,

Timo Eppig

Academic Editor

PLOS ONE

Additional Editor Comments (optional):

Reviewers' comments:

Reviewer's Responses to Questions

**Comments to the Author**

1. If the authors have adequately addressed your comments raised in a previous round of review and you feel that this manuscript is now acceptable for publication, you may indicate that here to bypass the “Comments to the Author” section, enter your conflict of interest statement in the “Confidential to Editor” section, and submit your "Accept" recommendation.

Reviewer #1: All comments have been addressed

Reviewer #2: All comments have been addressed

2. Is the manuscript technically sound, and do the data support the conclusions?

Reviewer #1: Yes

Reviewer #2: Yes

3. Has the statistical analysis been performed appropriately and rigorously?

Reviewer #1: N/A

Reviewer #2: Yes

4. Have the authors made all data underlying the findings in their manuscript fully available?

Reviewer #1: Yes

Reviewer #2: Yes

5. Is the manuscript presented in an intelligible fashion and written in standard English?

Reviewer #1: Yes

Reviewer #2: Yes

6. Review Comments to the Author

Reviewer #1: (No Response)

Reviewer #2: (No Response)

7. PLOS authors have the option to publish the peer review history of their article (what does this mean?). If published, this will include your full peer review and any attached files.

Reviewer #1: **Yes: **Ralf Blendowske

Reviewer #2: No

---

## [Editor Report · Acceptance letter]

PONE-D-25-24939R1

PLOS ONE

Dear Dr. Gatinel,

I'm pleased to inform you that your manuscript has been deemed suitable for publication in PLOS ONE. Congratulations! Your manuscript is now being handed over to our production team.

Kind regards,

on behalf of

Prof. Dr. Timo Eppig

Academic Editor

PLOS ONE